# Composite Uremic Load and Physical Performance in Hemodialysis Patients: A Cross-Sectional Study

**DOI:** 10.3390/toxins12020135

**Published:** 2020-02-22

**Authors:** Karsten Vanden Wyngaert, Amaryllis H. Van Craenenbroeck, Els Holvoet, Patrick Calders, Wim Van Biesen, Sunny Eloot

**Affiliations:** 1Department of Rehabilitation Sciences, Faculty of Medicine and Health Sciences, Ghent University, B-9000 Ghent, Belgium; patrick.calders@ugent.be; 2Laboratory of Experimental Medicine and Pediatrics, University of Antwerp, B-2650 Antwerp, Belgium; amaryllis.vancraenenbroeck@uantwerpen.be; 3Department of Nephrology, University Hospitals Leuven, B-3001 Leuven, Belgium; 4Department of Internal Medicine, Renal Division, Ghent University Hospital, B-9000 Ghent, Belgium; els.holvoet@uzgent.be (E.H.); wim.vanbiesen@uzgent.be (W.V.B.); sunny.eloot@uzgent.be (S.E.)

**Keywords:** end-stage renal disease, uremic toxins, muscle strength, exercise capacity, coordination and balance, physical performance, hemodialysis

## Abstract

Impaired physical performance is common in patients on hemodialysis (HD) and is associated with poor prognosis. A patient relevant marker of adequacy of dialysis is lacking. Previous studies evaluated uremic toxicity by assessing the impact of different uremic toxins separately. However, such an approach is most likely not reflective of true uremic toxicity. Therefore, this cross-sectional study aimed to examine if the uremic syndrome, estimated as one composite of different uremic toxins (facilitated by ridge regression method) to reflect the kinetic behavior during dialysis, is associated with physical performance in patients on HD. Levels of p-cresyl glucuronide and sulfate, indole-acetic acid, indoxyl sulfate, uric acid, hippuric acid, and 3-carboxy-4-methyl-5-propyl-2-furanpropionic acid were assessed and associated by ridge regression to muscle strength, functional exercise capacity, and measures of balance and coordination. 75 HD patients were included (mean age 68 years, 57% male). The composite of different uremic toxins (i.e., uremic load) explained 22% of the variance in handgrip strength. Although there was an association between full body muscle strength and the composite uremic load independent of nutritional status, age and gender, the predictive power of composite uremic load for muscle weakness is limited. Single uremic toxins as well as composite uremic load were not associated with exercise capacity, coordination, and balance, indicating that the degree of uremia does not predict physical performance in patients on HD.

## 1. Introduction

Over the last years, evidence has been compiled that the classical approach to the evaluation of dialysis adequacy by, e.g., Kt/V_urea_ might not be suitable [1]. First, the kinetic behavior of single uremic toxins (UTs), such as urea or creatinine is not representative of that of other toxins such as middle-sized molecules (representative example beta-2 microglobulin) and protein-bound toxins (representative examples p-Cresyl sulfate (pCS), p-Cresyl glucuronide (pCG), indoxyl sulfate (IS)) [2], and not even of that of other small soluble solutes [3]. Therefore, uremic toxicity as assessed by the impact of different uremic toxins separately (i.e., as individual contributors), most likely does not reflect true uremic toxicity. Furthermore, the toxicity of the individual molecules is probably more synergistic rather than just additive. When assessing dialysis adequacy, the uremic load represented as one composite outcome built up of a set of representative uremic toxins rather than single UTs need to be taken into account. Although a measure or equation as such does not exist, composite uremic toxicity can be represented by using theoretical constructs that incorporate multicollinearity into the model. By using these models, a conceptual association between composite uremic load and other outcomes can be examined. Second, several initiatives have emerged pointing to the necessity to use patient relevant rather than pure surrogate (biochemical) markers as outcomes in studies [4,5]. Survival is of course important, but it does not allow to assess the impact of alternative strategies in a short time frame. More ideal would be a marker that is associated with mortality, and that potentially can change already on relatively short-term with alternative dialysis strategies. For the moment, such a patient relevant, holistic marker is lacking in the field of dialysis adequacy. 

Physical performance is important to patients, as it impacts quality of life and is associated with mortality and morbidity [6]. Therefore, it might be one of the appropriate markers to be included in a holistic assessment of the quality of dialysis and of dialysis adequacy, provided there is a good association of physical performance with uremic load. Physical performance is an umbrella term that covers all physical aspects involved in the execution of daily activities. It can be assessed by *maximal measures* that determine the highest muscle force production (muscle strength) or the maximum amount of physical exertion (exercise capacity) and by *functional measures* that examine the performance of daily-encountered activities such as standing up (functional muscle strength) and walking (functional exercise capacity) [7]. Furthermore, poor body coordination and balance determine physical performance as well and can hamper complex daily activities.

In uremia, impairments in physical performance are common and associated with malnutrition and comorbidities such as cardiovascular (CV) disease and neuropathy. As various UTs promote these comorbidities [8], they most likely affect physical performance as well. However, the association between composite uremic load and decreased physical performance in patients on hemodialysis (HD) has thus far not been investigated in detail. Only one study, including 90 prevalent HD patients, identified serum asymmetric dimethylarginine concentration as a determinant of muscle strength, but not of exercise capacity [9].

Creatinine itself is not a suitable marker to assess the association between uremia and muscle strength, as high values can be either indicative of low clearance or of high muscle mass.

Hence, the present study was designed to examine the association between the composite uremic load by representative UTs and maximal and functional measures of physical performance including muscle strength, functional exercise capacity, and assessment of coordination and balance in patients on HD.

## 2. Results

### 2.1. Baseline Data and Demographics

Seventy-five patients were included in this study and baseline characteristics are described in Table 1. The cohort is representative for a population on HD (aged 68 years, 57% male and 16 months on dialysis). Muscle weakness (91%) and impaired functional exercise capacity (87%) were highly prevalent. According to the Davies CV morbidity classification, 48% and 39% of our population was identified with a medium and high mortality risk. Additionally, 47% and 27% of this cohort had diabetes and symptoms of neuropathy respectively. The uremic characteristics of our population are reported in Table 2.

### 2.2. Uremic Load and Physical Performance

The association between composite uremic load and different assessments of physical performance by ridge regression analysis is reported in Table 3. The variation in composite uremic load explained 22% of the variance in handgrip strength. Although composite uremic load was related to both maximal and functional measures of muscle strength when controlled for age and gender, and when controlled for age, gender, and nutritional status, no associations were found between composite uremic load and other measures of physical performance.

### 2.3. Individual Uremic Toxins and Physical Performance

Based on a detailed analysis of the relevant models reported above, all protein-bound toxins contributed to the relationship between composite uremic load and muscle strength (Appendix A). Nevertheless, no difference in concentration of a single uremic marker was identified in the between-groups analysis of the STS- and handgrip strength-tertiles (Appendix A respectively). However, patients with good maximal quadriceps strength showed lower levels of indoxyl sulfate (IS) compared to those with moderate or poor quadriceps strength (Appendix A). 

### 2.4. Uremic Load and Prognosis as Based on a Functional Parameter

A logistic ridge regression controlled for age and gender showed that the degree of uremia is related to the functional parameter six-minute walking test (6MWT) higher or below 300 m, which is in itself associated with mortality and quality of life in patients on HD (Table 4). Composite uremic load rather than individual UTs determined performance-based prognosis in this cohort (Appendix A).

## 3. Discussion

The present study provides limited evidence of an association between physical performance and composite uremic load in 75 patients on HD. The variation in composite uremic load rather than specific UTs explained various aspects of muscle function, including upper limb and lower limb muscle strength. However, both single UTs and composite uremic load were not associated with other measures of physical performance such as exercise capacity, balance, and coordination.

The need to use patient relevant rather than biochemical surrogates as outcomes of clinical studies has recently gained a lot of attention [4,5]. Various studies aimed to identify a gold quantifier of uremic burden that is associated with hard endpoints such as Quality of Life (QoL), morbidity, and mortality [8]. As such, pCG, pCS, HA, and IS are pro-inflammatory metabolites and promote endothelial dysfunction, atherosclerosis, and anemia in patients with chronic kidney disease, resulting in an increased risk of cardiovascular events and mortality [10,11,12,13,14,15]. Next, CMPF and UA are associated with neurological abnormalities [16,17] and insulin resistance respectively [18]. Although this multifactorial association between UTs and hard endpoints is extensively described in literature, only a fraction of the approximately 130 UTs are identified as relevant and the association between composite uremic load and patient relevant outcome has not yet been examined. Furthermore, to our knowledge, no studies report on the association between QoL and uremic toxicity. Nevertheless, prognosis as assessed by 6MWT higher or below 300 m is associated with all the three endpoints (i.e., morbidity, mortality, and QoL) and, therefore, could act as that representative marker for the patients’ health and well-being [6]. We found that composite uremic load, but not specific UTs determine functional prognosis in patients on HD, albeit no association was found between uremic load and the 6MWT as continuous variable. This discrepancy indicates that composite uremic load may predict the patients’ health and well-being, but is not appropriate to predict functional exercise capacity in patients on HD. In line with other findings, our results indicate that patients treated with dialysis of high-quality and adequacy, and thus presenting with a lower uremic load, have a better prognosis, as based on a functional marker, compared to those with poor dialysis-quality [19,20,21].

It appears not unexpected to accept that mainly other factors related to dialysis play a role in the representative features of physical performance, such as nutritional status, level of physical activity or comorbidities. Patients on HD are subject to at least 12-h of physical inactivity per week and show poor levels of physical activity on both dialysis and non-dialysis days compared to healthy subjects and kidney transplant recipients [22]. Accordingly, HD itself promotes the vicious circle of physical inactivity and impaired exercise capacity [23]. Physical activity has been recognized as a major treatable aspect in patients with high CV risk and results in improved physical performance and prognosis [24,25]. Therefore, it should be encouraged as much as possible to all patients on HD.

In the present study, we observed a low association between maximal as well as functional measures of muscle strength and composite uremic load. It is important to notify that muscle strength is a part of physical performance, and, more importantly, that it is only partially explained by muscle mass. Moreover, in our study, the association between composite uremic load and muscle strength was only slightly affected by nutritional status as assessed by MNA. These observations dovetail with other findings that malnutrition and reduced muscle mass explain only to a limited extend the presence of muscle weakness [26]. Sato et al. examined the association between UTs and muscle characteristics, and report a toxic effect of IS on the proliferation and viability of skeletal muscle cells. In addition to these fiber-volume changes, IS and pCS alter energy production by the downregulation of mitochondrial function as well, resulting in an ATP-shortage in skeletal muscle cells [27,28]. As such, uremia-induced alterations in ATP bioavailability provide a non-muscle mass-related explanation for muscle weakness in patients on HD. Improvement of uremic load can thus potentially already have a positive impact on muscle strength in the short term, an effect that in the longer term might be strengthened by its impact on nutritional status. In contrast, the impact of improving composite uremic load might not unambiguously result in improved daily physical performance of the patient on HD as, even though our analysis is corrected for age, gender, and nutritional status, no associations were found between composite uremic load and exercise capacity, coordination, and balance.

The uremic syndrome is a multifactorial condition affecting the inflammatory, CV, and neural systems [8]. Comorbidities related to these systems, such as malnutrition, CV disease, neuropathy, and frailty also contribute to impaired physical performance in patients with end-stage kidney disease [29]. Next to solute removal, aspects of dialysis that contribute to malnutrition and inflammation will therefore have an impact on physical performance as well. Including the evaluation of functional prognosis and muscle strength in studies assessing quality of dialysis thus also seems logical from that perspective.

Similar to functional exercise capacity, the uremic syndrome may indirectly more than directly cause impairments in gait quality and balance. Diabetes, osteoporosis, and micro- and macro-vascular impairments are common in patients on HD and result in non-uremic neuropathy and musculoskeletal complications (e.g., amputations, fractures, and Charcot foot) [30,31]. Various complications in the lower limbs can lead to poor gait quality and balance, and promote an already increased risk of falls in this cohort [32]. In contrast to the impact on exercise capacity and muscle strength, no literature exists on the effect of rehabilitation on neurological outcome in patients on HD. 

Our study has some limitations. First, this study included 12 UTs for only 75 observations and could be underpowered. Nevertheless, despite the shrinkage estimator, we did find associations between composite uremic load and muscle strength, indicating that this marker could be representative to some extent. Second, only a fraction of all potential UTs was analyzed and included in our marker of composite uremic load, thus, the actual uremic load could still be different. However, we countered this limitation by including established and accepted representative UTs with different protein-bound features. Third, the composite uremic load, as discussed in this study, is only a conceptual outcome and cannot be quantified or used as an outcome measure in different patients. Fourth and final, researchers and patients were not blinded. 

## 4. Conclusions

In conclusion, composite uremic load rather than specific uremic toxins influences measures of muscle function in patients on hemodialysis. No associations were found between exercise capacity, coordination, and balance on the one hand and single UTs and composite uremic load on the other. Therefore, composite uremic load might not predict true daily physical functioning in patients on hemodialysis. It appears that mainly other factors related to dialysis contribute to the high prevalence of physical impairments in these patients.

## 5. Materials and Methods

### 5.1. Participants and Study Design

This multicenter cross-sectional study recruited prevalent HD patients between December 2016 and March 2018. Patients were eligible if older than 18 years, not pregnant, able to adequately respond to verbal commands, and willing to provide written informed consent. Patients who had significant surgical interventions in the preceding 6 months were excluded. The study complies with the Declaration of Helsinki, was approved by the Ghent and Antwerp ethics committees (project number Ghent EC B670201525559; 15 October 2015 and Antwerp EC B300201422642; 7 December 2016), and written informed consent was obtained from all participants. Details and full prescription of the protocol and tests are reported on clinicaltrial.gov (retrospective registration: NCT03910426).

Baseline demographic data were obtained from the medical records, and the Mini-Nutritional Assessment (MNA)-long form was used to assess nutritional status.

### 5.2. Physical Performance

Maximal and functional measures of physical performance were examined, and included (1) handheld dynamometry (Microfet; Biometrics, Almere, the Netherlands) of the upper and lower limb muscles (i.e., handgrip and quadriceps strength) for the maximal assessment and (2) five-repetition sit-to-stand test (STS, functional muscle strength) and six-minute walking test (6MWT, functional exercise capacity) for the functional assessment of physical performance. Furthermore, balance, coordination, and gait quality were examined by the Frailty and Injuries Cooperative Studies of Intervention Technique-4 (FICSIT) and the Tinetti test. The order of physical tests was randomized, and the tests were performed before dialysis, albeit the isometric measures of muscle strength (i.e., handgrip and quadriceps strength) were assessed during dialysis to reduce the time claimed from patients before dialysis.

The following limits of normal were used: 80% of the expected value for age, gender, and body mass index (BMI) for muscle strength and 6MWT [33,34,35], 15 s for STS [36], and 11/12 points for the Tinetti test [37]. Patients comparable to end-stage kidney disease (ESKD) patients were identified as having a poor prognosis as based on a functional measure of physical performance when they reached <300 m on the 6MWT [38].

A worst possible score was given to patients that failed to complete that specific physical test.

### 5.3. Uremic Toxins

Blood samples were collected just before the start of the midweek dialysis session from the patient’s vascular access.

Samples were centrifuged within 20–30 min after collection during 10 min at 1250 g and at 4 °C. Subsequently, the serum was stored at −80 °C until batch analysis. All UTs were analyzed at the Ghent University Hospital [2,39] and were determined by reversed-phase high performance liquid chromatography [40,41] using the following methods: (1) the fluorescence detection method for protein-bound toxins p-Cresyl glucuronide (pCG, molecular weight 284 Da), indole-acetic acid (IAA, 175 Da), indoxyl sulfate (IS, 213 Da), and p-Cresyl sulfate (pCS, 187 Da) and (2) the UV detection method for uric acid (UA, 168 Da), the protein-bound toxins hippuric acid (HA, 179 Da), and 3-carboxy-4-methyl-5-propyl-2-furanpropionic acid (CMPF, 240 Da). 

Total concentrations of protein-bound toxins were determined by consecutively 30 min deproteinizing by heat denaturation, 10 min cooling, 10 min centrifuging (7379 g), and by finally filtering the serum for 20 min (3615 g, Amicon Ultra 0.5 mL Filters, Millipore, Billerica, MA, USA). Free fractions of protein-bound toxins were obtained using a Centrifree^®^ filter device (Millipore Billerica, MA, USA). 

For cases where serum level was below the limit of detection, the lowest detectable value divided by 2 (i.e., IS, 0.00025 mg/dL; pCS, 0.003 mg/dL; pCG, 0.0012 mg/dL; IAA, 0.00005 mg/dL; CMPF, 0.0001 mg/dL; HA, 0.006 mg/dL; UA, 0.0006 mg/dL) was implemented.

Composite uremic load was defined as one indicator of uremic toxicity estimated by clustering the above reported uremic toxins in a regression model.

### 5.4. Statistical Analysis

Based on the distribution, baseline data were expressed as mean (SD), median [IQR], or frequency (n; %). Between groups analysis was performed by the one-way ANOVA or Kruskal-Wallis; the Scheffe’s or Mann-Whitney U tests were used in the post hoc comparisons, respectively. Least Absolute Shrinkage and Selection Operator (LASSO) and ridge regression methods were used to examine the association between the composite uremic load and different domains of physical performance. These methods were used to examine data that suffer from multicollinearity by L1 (LASSO) and L2 (ridge) regularization techniques. Accordingly, a univariate association was examined between one dependent variable (e.g., muscle strength) and a cluster of collinear independent variables (e.g., composite uremic load by several individual UTs) without overestimating the association, as would be the case in general linear models suffering from multicollinearity. In other words, by using these models the global impact of uremic toxicity (i.e., composite uremic load represented by different single UTs) on the different measures of physical performance could be examined. The collinearity penalty implies that at least one UT should contribute to the overall effect size as a precondition for an association with composite uremic load. Mean-squared errors (MSE) were calculated for both methods and the method with the lowest MSE was selected (in this case, ridge). Ridge regressions were performed using the lmridge package in R and the general cross-validation method was used to estimate the optimal k [42]. Patients were allocated to one of three groups based on the tertiles of physical tests and between groups analysis was performed using the Kruskal-Wallis and Mann-Whitney U test to examine the associations of single UTs with the different measures of physical performance. The alpha-level was set at *p* < 0.05 and in contrast to the ridge regressions, all analyses were performed using IBM Statistical Package for Social Sciences (SPSS version 25, IBM Corp., Armonk, NY, USA).

## Figures and Tables

**Table 1 toxins-12-00135-t001:** Patient characteristics.

Variable	Total
	(*n* = 75)
Age (years)	68.0 ± 15.3
Male (*n*; %)	43; 57.3
BMI (kg/m^2^)	26.1 ± 5.1
Dialysis vintage (months)	16.0 [9.0; 39.0]
Type of dialysis	
Hemodialysis (*n*; %)	31; 41.3
Hemodiafiltration (*n*; %)	44; 58.7
Medication taken on a daily basis (*n*)	13.0 ± 3.9
Davies comorbidity score (0–7)	2.0 ± 1.3
Comorbidities	
Diabetes (*n*; %)	35; 46.7
CVD (*n*; %)	54; 72.0
Neuropathy (*n*; %)	20; 26.7
Retinopathy (*n*; %)	24; 32.0
Respiratory disorders (*n*; %)	21; 28.0
Musculoskeletal disorders (*n*; %)	28; 37.3
Etiology of chronic kidney disease	
Glomerulonephritis (*n*; %)	12; 16.0
Hematologic malignancies (*n*; %)	2; 2.7
Interstitial nephropathy (*n*; %)	7; 9.3
Diabetic nephropathy (*n*; %)	19; 25.3
Hypertension, angiosclerosis or unknown (*n*; %)	31; 41.4
ADPKD (*n*; %)	4; 5.3
Hemoglobin (g/dL)	11.1 ± 1.5
CRP (mg/L)	3.6 [2.5; 10.0]
Total serum protein (g/L)	65.0 ± 6.2
Anion gap (mEq/L)	12.0 [11.0; 19.1]
Quadriceps force (N)	186 ± 73
Patients with pathological quadriceps force (*n*; %)	68; 90.7
Handgrip force (kg)	30.0 ± 11.1
Patients with pathological handgrip force (*n*; %)	20; 26.7
Tinetti (/12)	11.0 [6.0; 12.0]
Patients at increased risk of falls (*n*; %)	35; 46.7
Sit-to-Stand (s)	18.0 [12.0; 50.0]
Patients at increased risk of falls (*n*; %)	47; 62.7
FICSIT (/28)	16.0 [10.0; 22.0]
6MWT (meters)	255 [110; 420]
Patients with pathological 6MWT (*n*; %)	65; 86.7
Patients scoring < 300 m on 6MWT (*n*; %)	43; 57.3

Values are expressed as mean ± SD, median [interquartile range] or as frequency (number; percentage). Abbreviations: ADPKD, autosomal dominant polycystic kidney disease; BMI, body mass index; CRP, C-reactive protein, CVD, cardiovascular disease; FICSIT, Frailty and Injuries Cooperative Studies of Intervention Technique-4; 6MWT, six-minute walking test.

**Table 2 toxins-12-00135-t002:** Uremic characteristics.

Variable (mg/dL)	Total Levels (*n* = 75)	Free Levels (*n* = 75)
IS	1.435 [0.844; 2.367]	0.092 [0.031; 0.166]
pCS	3.478 [2.303; 4.518]	0.212 [0.116; 0.369]
pCG	0.191 [0.066; 0.502]	0.166 [0.062; 0.449]
IAA	0.119 [0.081; 0.188]	0.037 [0.016; 0.059]
HA	1.820 [0.634; 3.521]	0.993 [0.247; 1.707]
CMPF	0.411 [0.224; 0.886]	/
UA	6.106 [5.381; 7.016]	/

Data are presented as median [IQR]. Abbreviations: CMPF, 3-carboxy-4-methyl-5-propyl-2-furanpropionic acid; HA, hippuric acid; IAA, indole-acetic acid; IS, indoxyl sulfate; pCG, p-cresyl glucuronide; pCS, p-cresyl sulfate; UA, uric acid.

**Table 3 toxins-12-00135-t003:** The association between physical performance and composite uremic load.

Outcome	Model 1 (Composite Uremic Load)		Model 2 (Model 1 + Age + Gender)		Model 3 (Model 1 + Age + Gender + MNA)		Model 4 (Age + Gender + MNA)	
	R^2^	*p*-Value	R^2^	*p*-Value	R^2^	*p*-Value	R^2^	*p*-Value
Quadriceps strength	0.101	0.268	0.421	<0.001 *	0.493	<0.001 *	0.386	<0.001
Handgrip strength	0.220	0.046 *	0.487	<0.001 *	0.650	<0.001 *	0.551	<0.001
Sit-to-Stand	0.163	0.119	0.331	0.002 *	0.389	0.001 *	0.211	<0.001
6MWT	0.142	0.280	0.333	0.002	0.401	0.002	0.282	<0.001
Tinetti	0.112	0.432	0.227	0.090	0.259	0.087	0.125	0.006
FICSIT	0.127	0.381	0.357	0.002	0.393	0.003	0.284	<0.001

Data represent the fit of ridge regression models to the measures of physical performance and are reported as R squared. The following variables were introduced in ridge regression Model 1: total and free concentrations of IS, pCS, pCG, IAA, HA, CMPF, and UA; gender and age were added resulting in Model 2 and gender, age, and MNA in Model 3; only age, gender, and MNA were introduced in model 4; * Association between composite uremic load and measure of physical performance (i.e., ≥1 UTs contributed to the overall effect size (*p* < 0.05)). Abbreviations: UTs, uremic toxins; 6MWT, six-minute walking test; MNA, mini-nutritional assessment scale.

**Table 4 toxins-12-00135-t004:** Logistic ridge regression of composite uremic load with performance-based prognosis.

Composite Uremic Load	Prognosis as Based on 6MWT (<300 m)	
	R Squared	*p*-Value
Model Fit	0.367	0.003
**Detailed Analysis of Composite Uremic Load**	**Estimate (SE)**	***p*-Value**
IS total	−0.38 (0.8)	0.639
IS free	0.27 (1.3)	0.837
pCS total	−2.03 (0.8)	0.017
pCS free	1.75 (1.2)	0.163
pCG total	2.33 (1.2)	0.048
pCG free	−1.21 (1.2)	0.318
IAA total	−0.720 (1.3)	0.571
IAA free	−0.61 (1.7)	0.715
HA total	1.03 (1.5)	0.496
HA free	−0.79 (1.6)	0.629
CMPF	−0.09 (0.5)	0.834
UA	0.05 (0.5)	0.908

The R squared value represents the fit of the ridge regression model to prognosis based on the 6MWT as surrogate measure. The contributions of different UTs to the model are reported in the detailed analysis and are presented as estimated beta-values and estimated standard error (SE); the analysis is controlled for age and gender. Abbreviations: 6MWT, six-minute walking test.

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
