# Peer review of "Composite Uremic Load and Physical Performance in Hemodialysis Patients: A Cross-Sectional Study"

_toxins, 2020, doi:10.3390/toxins12020135_

Round 1
Reviewer 1 Report
The authors examined the association between physical performance and uremic toxins in hemodialysis patients with a cross-sectional study. There are few reports to discuss uremic toxins and physical functions in clinical studies, and this study is important. ‘Total uremic load ‘is a new concept to understand the effect of uremia on the various systemic disorder in chronic kidney disease patients while the authors should explain it in detail.
It is very hard for readers to understand the meaning of total uremic load. The authors should explain it enough in Abstract and Methods. The authors should explain the meaning of ‘Daily drug use’ in Table 1. The authors should put actual data of serum level of each uremic toxin as well as total uremic load in Table 1 Introduction: line 62-65 should be discarded. Table 2 showed ’The association between physical performance and total uremic load’ while covariates of Model 1 included each UTs. It may be associated with the meaning of ‘total uremic load’ and the authors should explain it in detail. Table 3 showed ’Logistic ridge regression of total uremic load with performance-based prognosis.’ while it showed the association between each UTs and 6MWT. It may be associated with the meaning of ‘total uremic load’ and the authors should explain it in detail.
Reviewer 2 Report
I agree with you that the more UTs increases in plasma, the more frailty develops. And the estimation for the severity of frail should be done with multiple factors. However, the manuscript is still lacking some data to explain this issue. Please answer my questions below and revise the manuscript.
#1. Please present the comorbid disease of patients (e.g., IgA nephropathy, Diabetic nephropathy)
#2. Please submit the variance of comorbidities, especially past cardiovascular disease, malignancy, infections, musculoskeletal disorders.
#3. Please summarize each uremic toxin concentration, and please show the summary of the total uremic load.
#4. Also, if available, please summarize the plasma concentration of known kidney function parameters (e.g., creatine, urea nitrogen), as well as the plasma concentration of albumin and hematocrit or hemoglobin.
#5. Additionally, please stratify the baseline characteristics data of #1 to #4 by high total uremic toxin group and low total uremic toxin group.
#6. The definition of the total uremic load is vague. Please describe more precisely the definition (e.g., calculation formula and units), and please compare the value between total uremic load and a specific uremic toxin. If it means the inclusion of all UTs as variables, please explain why you conclude there is strength in total uremic load though Model 1 has low R-squared with limited statistical significance.
#7. I assume you chose the ridge regression model to increase the number the variables rather than using a linear regression model. But I am concerned about the collinearity of variables such as total and free UTs. Can you show me the VIF among each variable?
Additionally, though you used the ridge regression analysis, it seems overfitting for your study population. It has just only 75 cases, but you include 14 variables at least in your model.
Please make a proof of validity in statistics from an expert or reconsider the statistic analysis.
Reviewer 3 Report
The manuscript is a well-designed study addressing the relationship between several uremic toxins and physical performance.
Few remarks:
Its unclear whether confounding factors have been taken in account. I would better characterize the cohort, focusing on parameters that may somehow affect physical activity, for example plasma hemoglobin, albumin levels, BMI. Also inflammation molecules may help. According to the method section, the tests were done before or during dialysis. It is important to be clear on this point, as dialysis per sè may impact the result. In addition, it is unclear whether the study is a blind study. Please clarify the type of dialysis and if it was the same in all enrolled patients. I would expand the discussion, as in the current version it is quite poor. There are several studies in the literature focusing on the possible role of single uremic toxins as detrimental factors contributing to the augmented mortality and just a few studies focusing on the 'toxins load''. I would start from this evidence.
Round 2
Reviewer 1 Report
The authors have revised their manuscript according to the suggestions from the editor and reviewers. Since the concept of 'total uremic load' is a new concept, the description of it in the methods and results is still hard for readers to understand.
The authors should show actual data of 'total uremic load' while they showed only conceptional methods. In relation to comment 1, they should show the data of each uremic toxins including indoxyl sulfate and p-cresyl sulfate in Table 1, not supplemental data.Author Response
Please see the attachment.

Reviewer 2 Report
In the revised version of the manuscript and the responses to our comments have made the study more comprehensive to the readers. But the fundamental point, the meaning of "total uremic load," is still unclear and hard to understand. If this parameter is a product of ridge regression, I assume there must be a formula made by the analysis results and the patient's uremic toxin level.
As long as the meaning of total uremic load cannot be understandable to all readers, the manuscript cannot be acceptable to be published. Please describe it more precisely.
Also, if you would like to emphasize the importance of total uremic load, you should perform the ridge regression analysis in Table 2 by comparing the model of adjusted only by age, gender, and MNA. This would be helpful for the readers to recognize adding the total uremic load predicts the performance of the physical activity more accurately rather than other parameters.
Reviewer 3 Report
The revised manuscript has addresses all referees' comments. Several additional information have been provided.
In my opinion the entire story reads well. There are some interesting data that require further confirmation in a larger population, but that are a nice 'starting points' for future studies.
Round 3
Reviewer 1 Report
The authors have responded to the reviewer's comments.
Author Response
We would like to thank the reviewer for his/her constructive remarks
Reviewer 2 Report
Comments to Authors,
I appreciate your understanding and making correction based on our comments. I am quite understanding what you would like to express from this manuscript to the readers. However, the results and the comprehension of this result are quite confusing. Please make a deep reconsideration on your manuscript by regarding my requests;
#1 In the new Table 2, you described the R-square and p-value of model 4. According to this result, the R-square of model 4 seems to be larger than model 1, and most physical examination variables were statistically significant by the ridge regression model in model 4. This comprehension should be "model 4 predicts the physical examination better than model 1", and "adding composite UT to the model 4 (model 3)" worsen the prediction. This might mean that the composite UTs "does not predict" the physical activities. I think this comprehension is more clear to the readers.
Concerning the results, I think you should not hesitate to give negative results. Since it is worthwhile if we can reduce expensive UT examination to predict the outcome of the physical activity. These results can be meaningful data if you change the comprehension vice versa.
The feasibility of this study is that single and also composite UTs do not affect the physical activity of dialysis patients. The researchers in the world believed that the UTs will affect the physical activity of dialysis patients, but the authors have discovered that it doesn't.
#2 If you change the comprehension of the results, I recommend to change the content of the abstract.
